# Around-the-Clock Noise Induces AD-like Neuropathology by Disrupting Autophagy Flux Homeostasis

**DOI:** 10.3390/cells11172742

**Published:** 2022-09-02

**Authors:** Pengfang Zheng, Xiaojun She, Chunping Wang, Yingwen Zhu, Bo Fu, Kefeng Ma, Honglian Yang, Xiujie Gao, Xiaofang Li, Fangshan Wu, Bo Cui

**Affiliations:** 1School of Public Health, Weifang Medical University, No. 7166, Baotong Xi Street, Weicheng District, Weifang 261053, China; 2Tianjin Institute of Environmental and Operational Medicine, No. 1, Dali Road, Heping District, Tianjin 300050, China; 3School of Public Health and Management, Binzhou Medical University, No. 346, Guanhai Road, Laishan District, Yantai 264003, China

**Keywords:** around-the-clock noise, AMPK-mTOR, autophagosome, lysosome, Alzheimer’s disease

## Abstract

Environmental noise is a common hazard in military operations. Military service members during long operations are often exposed to around-the-clock noise and suffer massive emotional and cognitive dysfunction related to an Alzheimer’s disease (AD)-like neuropathology. It is essential to clarify the mechanisms underlying the effects of around-the-clock noise exposure on the central nervous system. Here, Wistar rats were continuously exposed to white noise (95 dB during the on-duty phase [8:00–16:00] and 75 dB during the off-duty phase (16:00–8:00 the next day)) for 40 days. The levels of phosphorylated tau, amyloid-β (Aβ), and neuroinflammation in the cortex and hippocampus were assessed and autophagosome (AP) aggregation was observed by transmission electron microscopy. Dyshomeostasis of autophagic flux resulting from around-the-clock noise exposure was assessed at different stages to investigate the potential pathological mechanisms. Around-the-clock noise significantly increased Aβ peptide, tau phosphorylation at Ser396 and Ser404, and neuroinflammation. Moreover, the AMPK-mTOR signaling pathway was depressed in the cortex and the hippocampus of rats exposed to around-the-clock noise. Consequently, autophagosome–lysosome fusion was deterred and resulted in AP accumulation. Our results indicate that around-the-clock noise exposure has detrimental influences on autophagic flux homeostasis and may be associated with AD-like neuropathology in the cortex and the hippocampus.

## 1. Introduction

Despite attempts to minimize sources of noise, hazardous noise exposure remains prevalent. Military service members are often appointed to long operations and live in environments that offer little time for recovery from hazardous noise [1,2,3]. They are exposed to various noise sources for durations over 12 h, and their average 24-h equivalent continuous sound levels (24-h) range from 71 to 127 dBA [1]. In addition, long-term continuous noise exposure can cause neurobehavioral dysfunctions, ultimately affecting their ability to perform cognitive-related tasks and missions [4,5,6]. However, the detailed mechanisms by which around-the-clock noise affects cognition are still unclear.

Our previous studies modeled noise exposure by simulating an industrial work environment with a duration of 4 h/day and showed that noise exposure is associated with cognitive decline as well as irreversible Alzheimer’s disease (AD)-like pathologies, such as hyperphosphorylated tau protein, amyloid-β (Aβ) accumulation, neuroinflammatory changes, and even neuronal apoptosis [7,8,9,10]. Here, we aimed to reveal whether the around-the-clock noise also causes AD.

Autophagy is a three-step process that includes induction, autophagosome (AP) formation, autophagosome–lysosome (AL) fusion, and degradation. Any abnormal step of autophagy can directly or indirectly lead to Aβ generation and tau aggregation, and increased tau phosphorylation aggravates neuronal cell variation and accelerates the course of AD [11,12,13,14,15]. Autophagy flux, defined as the rate of degradation of long-lived protein aggregates by autophagy, involves the whole autophagy process. 

A hypoxic environment can cause abnormal mitochondrial energy production, calcium overload, and excessive production of reactive oxygen species, leading to oxidative stress, autophagy, and apoptosis of neurons and cardiomyocytes [16,17]. Our previous study showed that chronic noise exposure influences the mTOR and light chain 3 B (LC3B) [18]. We, therefore, hypothesized that the inhibited AMPK-mTOR signaling pathway induced by around-the-clock noise would disrupt the homeostasis of autophagy flux. We further aimed to examine whether these effects would subsequently derange degradation pathways to induce AD-like pathology.

This research intended to explore the mechanism of AD-like changes caused by exposure to around-the-clock noise. Based on our previous results, we hypothesized that the impairment of cognitive function seen in workers exposed to around-the-clock noise may be related to AD-like changes in the central nervous system, in which autophagy might play a key role. To understand the pathogenesis of AD, it is crucial to examine the dynamic changes in autophagic flux that occur over the course of AD. Consequently, we explored autophagy flux at different stages to acquire a better understanding of the underlying pathological mechanisms.

## 2. Materials and Methods

### 2.1. Animals and Experimental Groups

Twenty male Wistar rats were randomly divided into a control group and a 24-h noise exposure group. The Wistar rats were obtained from the Beijing Viton Lihua Laboratory Animal Technology (Beijing, China). All rats were housed under normal conditions with an environmental temperature of 23 ± 2 °C. Prior to the experiment, the rats underwent six days of adaptation. The rats had free access to standard laboratory rodent food and water. After the experiment, all animals were sacrificed under brief anesthesia for subsequent biochemical analyses. All experiments were performed in accordance with approved guidelines specified by the Animal and Human Use in Research Committee of the Tianjin Institute of Health and Environmental Medicine and the Three Rs principle.

### 2.2. Noise Exposure Set-Up

To simulate the noise intensity and exposure pattern of a military operational environment, the noise exposure conditions were set to 95 dB white noise during the day 8 h/d (8:00–16:00) and 75 dB white noise at night 16 h/d (16:00–8:00 the next day) for 40 days. A noise generator (BK 3560 C, B&K Instruments, Nærum, Denmark) was used to generate white noise, which was amplified using a power amplifier and broadcast through a loudspeaker inside a soundproof room. The animals were housed in individual wired cages for 95 dB noise exposure. The sound levels inside the cages were measured every hour. The difference in noise levels between different cages was less than 2 dB.

### 2.3. Western Blot Analysis

The frontal cortex and the hippocampus were sonicated in a high-throughput tissue grinder (Xinzhi Biotechnology, Ningbo, China) for 1 min. Lysates were centrifuged at 12,000× *g* at 4 °C for 10 min to obtain the supernatant fraction, which was used for Western blot analysis. The following primary antibodies were used: mouse anti-β Amyloid (B4) (sc-28365 1:1000, Santa Crutz Biotechnology, Dallas, TX, USA), rabbit anti-Tau (phospho S396) antibody (ab109390, 1:10,000, Abcam, Cambridge, UK), rabbit anti-Tau (phospho S404) antibody (ab109390, 1:10,000, Abcam, Cambridge, UK), mouse GFAP (Ser38) (AP0227 1:1000, Bioworld Technology, Louis Park, MN, USA), rabbit anti-P-AMPK-Thr172 antibody (2535, 1:1000, Cell Signaling Technologies, Danvers, MA, USA), rabbit anti-AMPK antibody (5832, 1:1000, Cell Signaling Technologies, Danvers, MA, USA), rabbit anti-mTOR antibody (P2476, 1:1000, Bioworld Technology, Louis Park, MN, USA), rabbit anti-p-mTOR-(Ser2448) antibody (5536, 1:1000, Cell Signaling Technologies, Danvers, MA, USA), rabbit anti-ULK1 antibody (ab240916, 1:1000, Abcam, Cambridge, UK), rabbit anti-LC3B antibody (ab192890, 1:2000, Abcam, Cambridge, UK), rabbit anti-SQSTM1\P62 antibody (AP6006, 1:1000, Bioworld Technology, Louis Park, MN, USA), rabbit anti-LAMP1 antibody (ab13523, 1:1000, Abcam, Cambridge, UK), rabbit anti-Cathepsin B antibody (31718, 1:1000, Cell Signaling Technologies, Danvers, MA, USA), and mouse anti-GAPDH antibody (MB001, 1:40,000, Bioworld Technology, Louis Park, MN, USA). The secondary antibodies used were anti-mouse/rabbit IgG and HRP-linked antibodies (BS12478/BS14278, 1:40,000, Bioworld Technology, Louis Park, MN, USA). Blots were quantified using an image analyzer (Complex2000, Nanjing Puoxin Biotechnology, Nanjing, China).

### 2.4. Real-Time Quantitative PCR (qPCR)

Following the manufacturer’s instructions, total RNA was extracted from the cortex and the hippocampus tissue using an RNeasy kit and then reverse transcribed into cDNA using a Transcriptor First Strand cDNA Synthesis Kit (TaKaRa Bio, Dalian, China). As described in Table 1, specific primers and probes for rat *BACE1*, *PS1*, *LC3B*, *CTSB*, *CTSD*, *IL6*, *NFκB*, *TNFα*, and *GAPDH* were used, *GAPDH* being the internal reference standard. In a 40-cycle PCR, the denaturation, annealing, and extension conditions of each PCR cycle were 95 °C for 30 s, 95 °C for 5 s, and 60 °C for 30 s, respectively. Real-time PCR was executed using on-demand gene expression assays and a Takara PCR thermal cycler dice real-time system (TaKaRa Bio, Dalian, China). After standardizing the Ct value to *GAPDH* expression, mRNA levels were calculated and presented as fold-induction values (2^−ΔΔCt^) relative to those of the control rats.

### 2.5. Transmission Electron Microscopy (TEM)

Three rats from each group were sacrificed. The hippocampus was obtained and fixed with 2.5% glutaraldehyde in cacodylate buffer, dehydrated with an ethanol gradient, made transparent with xylene, immersed in paraffin, and cooled to −20 °C. Thin sections (50 nm) of the tissue were put on copper grids for inspection with a TECNAI G 20 TWIN electron microscope (FEI, Hillsboro, OR, USA) and doubly stained with saturated aqueous solutions containing 2% uranium acetate and 2% lead citrate.

### 2.6. Statistical Analysis

The graphs show group means and standard deviations. Statistical analysis was implemented using the SPSS 22.0 software (SPSS Inc., Chicago, IL, USA) and GraphPad Prism 8 (Graphpad Software, San Diego, CA, USA). The Student’s *t*-test was used to determine the statistical significance. Differences were considered statistically significant when the *p* value was <0.05. 

## 3. Results

### 3.1. Around-the-Clock Noise Exposure Is Associated with AD-like Neuropathology in the Cortex and Hippocampus of Rats

To investigate the effects of around-the-clock noise exposure on AD-like neuropathology, we evaluated the degree of tau hyperphosphorylation at Ser396 and Ser404 residues, neuroinflammatory changes, and Aβ accumulation in the cortex and the hippocampus using Western blotting. We also examined the mRNA expression levels of *PS1* and *BACE1*. The expression of Aβ42, Ser396, Ser404, and *GFAP* in the rat cortex and hippocampus were significantly higher (*p* < 0.05) in the around-the-clock noise group (Figure 1A–E,H–L). Furthermore, the mRNA expression of *PS1* and *BACE1* increased following 40 days of 24-h noise exposure (Figure 1F,G,M,N). These data suggest that exposure to around-the-clock noise may aggravate the neuropathology characteristically seen in AD.

### 3.2. Around-the-Clock Noise Exposure Is Associated with Neuroinflammation in the Cortex and Hippocampus of Rats

To study the impact of around-the-clock noise exposure on brain inflammation, the mRNA expression of *IL6*, *TNFα*, and *NFκB* in the cortex and the hippocampus were examined (Figure 2A–F), which were significantly higher in the rats exposed to around-the-clock noise compared with the control rats. These results are highly suggestive of the occurrence of an inflammatory response in the brain of the rats exposed to around-the-clock noise.

### 3.3. Around-the-Clock Noise Exposure Suppresses the Initial Stage of Autophagy Driven by the AMPK-mTOR Pathway

Many signaling pathways are implicated in autophagy, including those involving mTOR and AMPK [19,20,21]. mTOR negatively regulates autophagy, whereas AMPK positively regulates it [22,23]. To elucidate the mechanism of around-the-clock noise-induced autophagy, we evaluated the levels of an mTOR target protein ULK1. As a Ser/Thr kinase, mTOR is known to inhibit autophagy by suppressing the phosphorylation of ULK1 [24]. The cortex and the hippocampus of the rats exposed to around-the-clock noise exhibited significantly higher (*p* < 0.05) expression of p-mTOR and mTOR (Figure 3A,E–G,K,L) compared with the control rats. ULK1, a downstream target protein of mTOR, is commonly used as a proxy measure of mTOR activity [25]. We discovered that the expression of ULK1 (Figure 3A,D,G,J) in the cortex and the hippocampus was markedly reduced (*p* < 0.05) in the around-the-clock noise exposure rats compared with the control group. AMPK also negatively controls mTOR activity [26]. The expression of p-AMPK and AMPK (Figure 3A–C,G–I) in the cortex and the hippocampus of the noise-exposed rats was markedly decreased (*p* < 0.05) compared with those of the control rats. These results suggest that around-the-clock noise inhibits autophagy in the brains of the 24-h noise-exposed rats by regulating the AMPK/mTOR pathway.

### 3.4. Around-the-Clock Noise Exposure Increases AP Formation

To investigate the impact of around-the-clock noise exposure on autophagosome formation, we tested the mRNA and protein levels of microtubule-associated LC3B in the cortex and the hippocampus. The around-the-clock noise exposure group had a higher expression of LC3B than the control group (Figure 4A,B,D–F,H). Of note, an increase in LC3B can indicate either activated autophagy or impaired LC3B degradation during autophagy progression [27]. To clarify this, we further tested the level of SQSTM1/P62, a well-known substrate degraded in autophagy. The P62 level was much higher in the 24-h noise exposure rats compared with the control group (Figure 4A,C,E,G), demonstrating that the clearance of P62 was inhibited in the 24-h noise exposure rats. 

We next examined the APs in the hippocampus of the 24-h noise exposure rats using TEM. APs were observed in the hippocampus of the 24-h noise exposure rats but were difficult to detect in the control group rats (Figure 4I,J). Taken together, these data indicate that exposure to around-the-clock noise leads to AP accumulation. 

### 3.5. Around-the-Clock Noise Exposure Impairs Autophagosome–Lysosome Fusion and Degradation 

To investigate the impact of around-the-clock noise exposure on lysosome biogenesis, the levels of LAMP1, a lysosomal structural protein, in the cortex and the hippocampus were analyzed by Western blot (Figure 5). The expression of LAMP1 and cathepsin B (CTSB) protein was significantly decreased (*p* < 0.05) in rats exposed to around-the-clock noise compared with the control rats (Figure 5A–C,F–H). CTSB and cathepsin D (CTSD), two major lysosomal hydrolytic enzymes, were detected in the cortex and the hippocampus of rats exposed to around-the-clock noise. The mRNA levels of *CTSB* and *CTSD* were significantly lower (*p* < 0.05) in the rats exposed to around-the-clock noise (Figure 5D,E,I,J). These results suggest that autophagosome–lysosome fusion and degradation are impaired in the brains of the rats exposed to around-the-clock noise.

## 4. Discussion

In this research, we verified that around-the-clock noise exposure can induce a string of functional autophagic flux disruptions and AD-like neuropathology in the cortex and the hippocampus of rats; these include abnormalities in AMPK-mTOR signaling and autophagosome–lysosome fusion, accumulation of APs related to amyloid pathology, tau hyperphosphorylation, and neuroinflammation. Moreover, we discovered that autophagy may play a role in aggravating the occurrence of AD-like neuropathology after around-the-clock noise exposure, which may provide a new dimension to the mechanisms of the effect of around-the-clock noise exposure on the nervous system.

Previous studies have shown that exposure to noise causes auditive system abnormalities, such as inducing threshold shift and damaging synapses formed by inner hair cells and spiral ganglion nerves [28,29]. Noise exposure for 4 h/d can also induce non-auditory effects, such as spatial memory and learning disorders, and significantly increase AD-like changes in the hippocampus and the cortex [8]. Occupational exposure limits for noise are typically 85 dBA as an 8-h time-weighted average, with the remaining 16 h of the day recommended for off-duty to let recovery from any temporary auditory and non-auditory effects that may have happened during the preceding work shift. Long-term continuous noise exposure patterns in long oceangoing voyages are one of the most obvious factors showing the effect of noise exposure [1,30], which may increase the risk of irreversible damage in the nervous system of navy personnel. Here, we focused on the possible harmful effects of around-the-clock noise exposure and found that around-the-clock noise not only increased Aβ expression, tau phosphorylation at Ser396 and Ser404, and neuroinflammation but also seemed to cause more significant AD-like pathological changes compared with part-time exposure [31,32]. Therefore, it is extremely important to understand the underlying mechanism of around-the-clock noise exposure on AD-like neuropathology.

Previous studies have found that AD-like pathological Aβ and hyperphosphorylated tau protein were degraded via the autophagic pathway [33], and the dysfunction of autophagy in different stages caused neurodegenerative diseases, including the dysfunction of AMPK-mTOR-mediated autophagy initiation [34], blocked APs degradation [35], and decreased autophagy–lysosome function [36]. Our aim was to study the effect of around-the-clock noise on different stages of autophagy flow; therefore, we firstly detected the effect of around-the-clock noise on the AMPK-mTOR signaling pathway. Autophagy is regulated primarily by mTOR and AMPK, and a key regulator of autophagosome formation is ULK1. AMPK promotes autophagy by phosphorylating ULK1, whereas mTOR inhibits autophagy by inhibiting ULK1 phosphorylation [37,38]. AMPK-mediated inhibition of ULK1 involves two mechanisms: direct inhibition of phosphorylated ULK1 by AMPK and indirect inhibition of ULK1 phosphorylation by activating mTOR phosphorylation [39,40,41]. In our study, AMPK activity was notably impaired, mTOR activation was higher, and the expression of ULK1 was lower in the hippocampus and the cortex of the rats exposed to around-the-clock noise. These results show that around-the-clock noise-induced dyshomeostasis of autophagy flux is mediated through AMPK’s indirect inhibition of ULK1 by activating mTOR.

Next, we investigated the stage of AP formation. LC3B is the most commonly used marker for APs [42] and the presence of lipidated LC3B may signal the formation of APs. Lysosomal hydrolases degrade LC3B after lipidation [43]. Furthermore, P62 aggregates are widely used as an indicator of autophagic degradation [42]. In this study, we found that the expression of LC3B and P62 in the cortex and the hippocampus increased significantly after around-the-clock noise exposure. Increased P62 indicates a decrease in lysosomal degradation function, thereby leading to the accumulation of LC3B. These results suggest that around-the-clock noise exposure impairs autophagic flux and induces the accumulation of APs. Our TEM results also show increased APs after around-the-clock noise exposure. Because APs are normally quickly transported to the cell body and degraded, they are scarcely found in normal nervous system tissue. Aβ, PS1, and BACE1 complexes present abundantly in the APs when the maturation and degradation of APs become obstructed, leading to AD-like pathological changes in AD cases and mouse models [44,45]. Consistent with our results, the expression of LC3B and P62 was shown to be greatly elevated following exposure to manganese. Interestingly, this resulted from the damage to the autolysosome function [35]. Therefore, we speculate that the accumulation of APs after around-the-clock noise exposure is caused by the decrease in the autolysosome function.

Autophagosome–lysosome degradation is the last stage of autophagy. After fusing with lysosomes, the autophagosome contents and inner membrane are broken down by lysosomal enzymes. LAMP1 is found in autophagic and lysosomal organelles and helps sustain the structural integrity of the lysosomal membrane, which is crucial for lysosomal stability and autophagy [46]. The lysosomal proteases CTSB and CTSD are crucial for maintaining cellular proteostasis by converting substrates for endocytosis, phagocytosis, and autophagy [47]. Consequently, to further illustrate the possible reasons for increased APs by around-the-clock noise, we explored the potential effect of around-the-clock noise in lysosomes by measuring changes in LAMP1, CTSB, and CTSD levels in the cortex and the hippocampus. We found that the expression of these three proteins decreased markedly after exposure to around-the-clock noise. Consistent with this observation, a study found that AP accumulation in AD may synergistically cause the destruction of the autophagy–lysosome system [36]. These results demonstrate that an impaired fusion of the autophagosome with lysosome leads to the accumulation of APs, which, in turn, blocks the degradation of Aβ and tau phosphorylation, resulting in AD-like neuropathology. Taken together, our results suggest a disruption in the fusion of APs and lysosomes or a reduction in the number and function of lysosomes, ultimately leading to dyshomeostasis of autophagy flux and AD-like neuropathology in rats exposed to around-the-clock noise. 

## 5. Conclusions

Exposure to around-the-clock noise, likely by disrupting the homeostasis of autophagy flux, aggravates amyloid and tau pathology in AD-related brain regions. Chronic uninterrupted noise exposure causes abnormal AMPK-mTOR signaling and prevents autophagosome–lysosome fusion, leading to the accumulation of APs related to amyloid pathology, tau hyperphosphorylation, and neuroinflammation. Consequently, APs accumulate in the cortex and the hippocampus in a manner resembling AD-like neuropathology, as shown in Figure 6. These discoveries broaden our understanding of the mechanisms associated with the pathology of cognitive impairment following exposure to around-the-clock noise. It also provides the basis for the mechanism of neurodegeneration caused by a specific environmental factor. Furthermore, the study of dynamic changes in autophagic flux in the pathogenesis of AD is an important approach to reveal the pathogenesis of AD, offers a theoretical basis for the assessment of health risks, and highlights the need for future work in harm prevention and control strategies for military operators exposed to around-the-clock noise during extended naval voyages.

## Figures and Tables

**Figure 1 cells-11-02742-f001:**
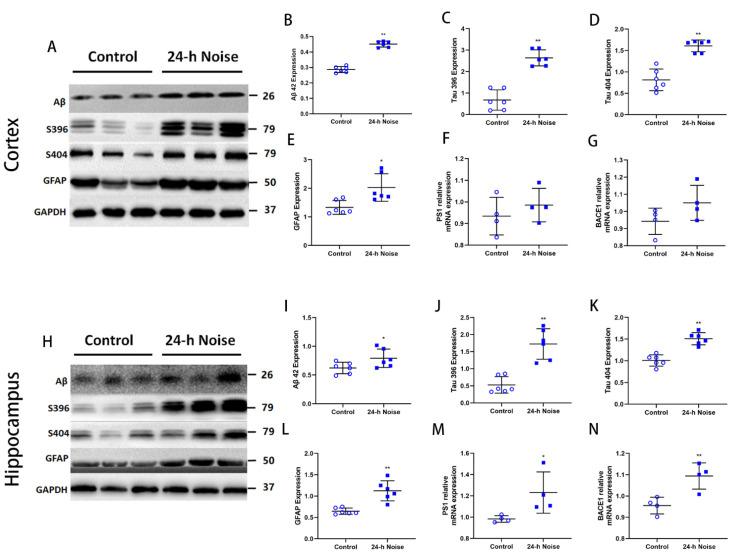
Around-the-clock noise (24 h/day) exposure leads to AD-like neuropathology in the cortex and hippocampus of rats. (**A**,**H**) Western blot of Aβ, phosphorylated tau, and GFAP expression in the cortex and hippocampus. (**B**–**E**,**I**–**L**) Quantification data showed that around-the-clock noise (24 h/day) increases the expression of Aβ, Tau396, Tau404, and GFAP in the cortex and hippocampus (*n* = 6 per group). (**F**,**G**,**M**,**N**) The mRNA expression of *PS1* and *BACE1* in the cortex and hippocampus (*n* = 4 per group). * *p* < 0.05, ** *p* < 0.01.

**Figure 2 cells-11-02742-f002:**
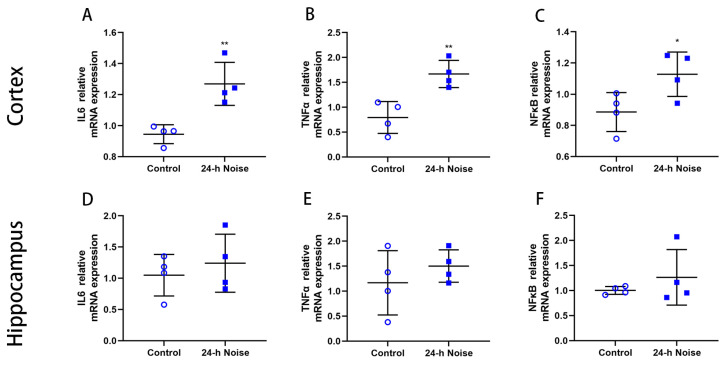
Around-the-clock noise (24 h/day) exposure is associated with neuroinflammation in the cortex and hippocampus of rats. (**A**–**F**) The mRNA expression of *IL6*, *TNFα*, and *NFκB* in the cortex and hippocampus (*n* = 4 per group). * *p* < 0.05, ** *p* < 0.01.

**Figure 3 cells-11-02742-f003:**
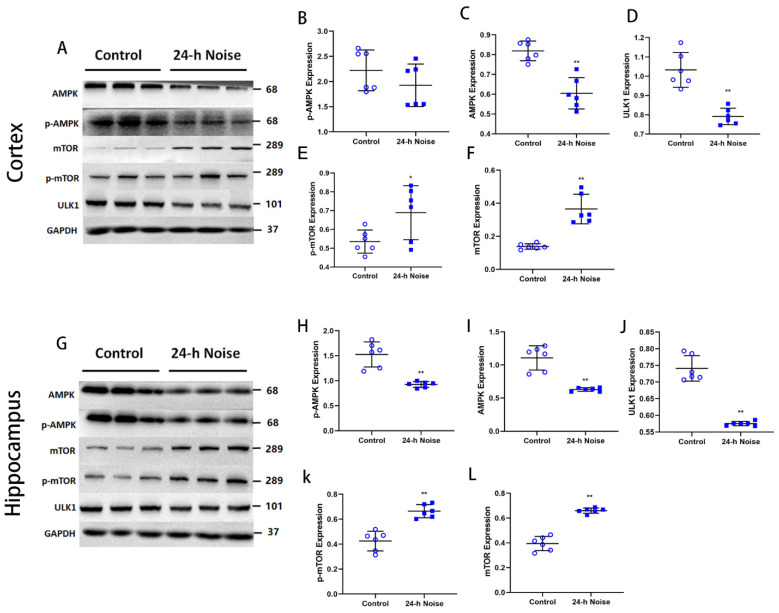
Around-the-clock noise (24 h/day) exposure suppresses the initial stage of autophagy driven by the AMPK-mTOR pathway. (**A**,**G**) Western blot of AMPK, p-AMPK, mTOR, p-mTOR, and ULK1 expression in the cortex and hippocampus. (**B**–**F**,**H**–**L**) Quantification data showed that around-the-clock noise (24 h/day) affects the expression of AMPK, p-AMPK, mTOR, p-mTOR, and ULK1 in the cortex and hippocampus (*n* = 6 per group). * *p* < 0.05, ** *p* < 0.01.

**Figure 4 cells-11-02742-f004:**
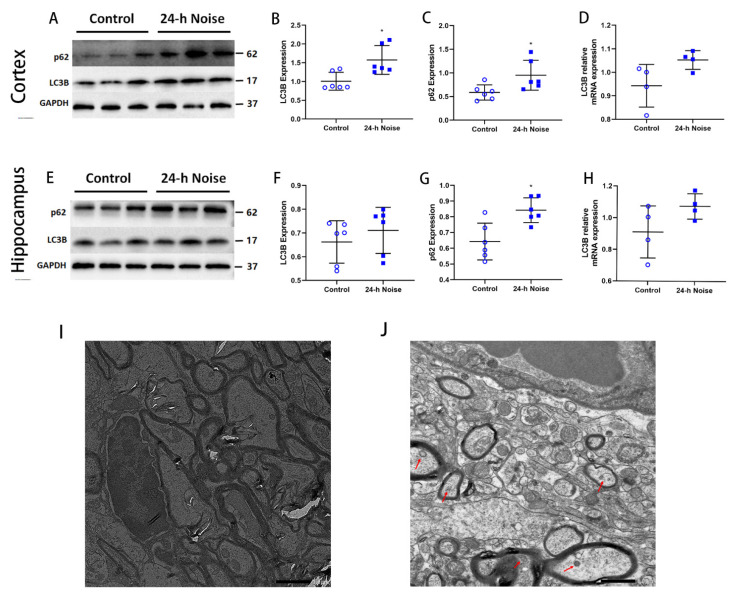
Around-the-clock noise (24 h/day) exposure increases AP formation. (**A**,**E**) Western blot of p62 and LC3B expression in the cortex and hippocampus. (**B**,**C**,**F**,**G**) Quantification data showed that around-the-clock noise (24 h/day) increases the expression of p62 and LC3B in the cortex and hippocampus (*n* = 6 per group). (**D**,**H**) The expression of *LC3B* at the RNA level in the cortex and hippocampus (*n* = 4 per group). (**I**,**J**) The ultrastructural hippocampus changes were observed with transmission electron microscopy (scale bar: 1μm). * *p* < 0.05.

**Figure 5 cells-11-02742-f005:**
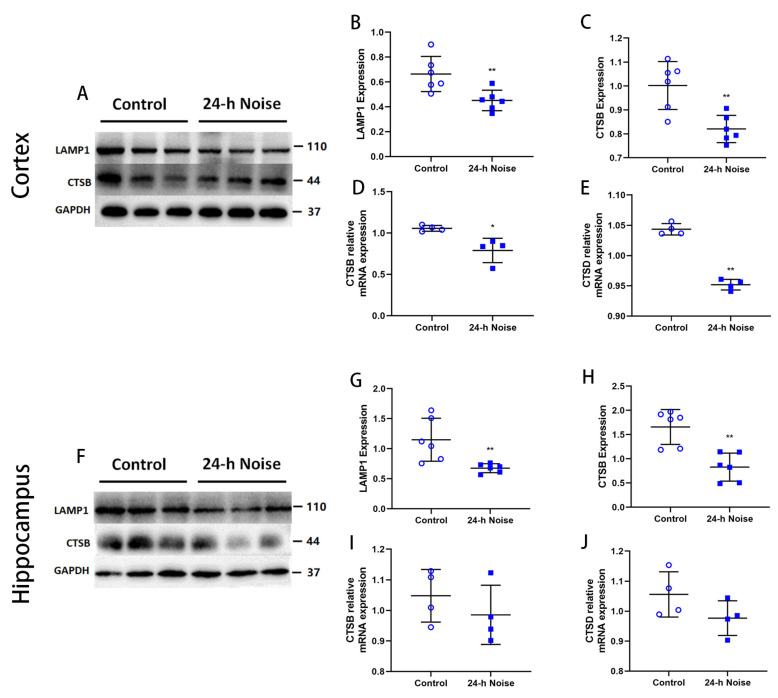
Around-the-clock noise (24 h/day) exposure impairs autophagosome–lysosome fusion and degradation. (**A**,**F**) Western blot of LAMP1 and CTSB expression in the cortex and hippocampus. (**B**,**C**,**G**,**H**) Quantification data showed that around-the-clock noise (24 h/day) decreases the expression of LAMP1 and CTSB in the cortex and hippocampus (*n* = 6 per group). (**D**,**E**,**I**,**J**) The mRNA expression of *CTSB* and *CTSD* in the cortex and hippocampus (*n* = 4 per group). * *p* < 0.05, ** *p* < 0.01.

**Figure 6 cells-11-02742-f006:**
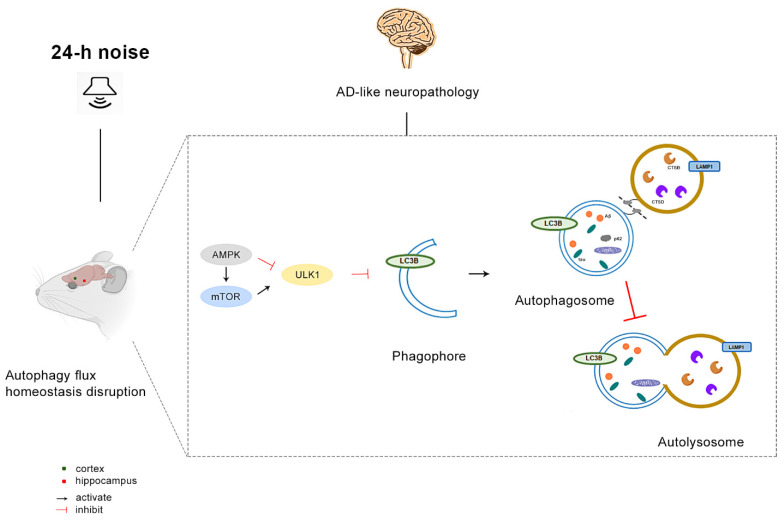
Around-the-clock noise (24 h/day) exposure disrupts autophagy flux, inducing AD-like neuropathology. Schematic representation of the findings of this study. Around-the-clock noise (24 h/day) exposure leads to AMPK inhibition, which then activates mTOR signaling to inhibit autophagy. Autophagy is impaired in the stages of autophagosome–lysosome fusion and degradation in rats exposed to around-the-clock noise. Inhibition of autophagy associated with around-the-clock noise exposure may aggravate the burden of APs and foster the development of AD-like neuropathology.

**Table 1 cells-11-02742-t001:** Rat primer sequences used for quantitative real-time PCR.

Gene	Primers
*BACE1*	F:5’-TCTGTCGGAGGGAGCATGAT-3’
	R:5’-GCAAACGAAGGTTGGTGGT-3’
*PS1*	F:5’-CATCATGATCAGTGTCATTGTTGT-3’
	R:5’-TGCATTATACTTGGAATTTTTGGA-3’
*LC3B*	F:5’-GGAAGATGTCCGGCTCATC-3’
	R:5’-CTTCTCACCCTTGTATCGCTCTAA-3’
*CTSB*	F:5’-AGGCTGGACGCAACTTCTAC-3’
	R:5’-ACTGTTCCCGTGCATCAAA-3’
*CTSD*	F:5’-CCTGGGCGATGTCTTTATTG-3’
	R:5’-GGCAAAGCCGACCCTATT-3’
*IL6*	F:5’-AGAGACTTCCAGCCAGTTGC-3’
	R:5’-TGAAGTCTCCTCTCCGGACT-3’
*NFκB*	F:5’-TGTCTGCACCTGTTCCAAAGAT-3’
	R:5’-TGCCAGGTCTGTGAACACTC-3’
*TNFα*	F:5’-CGTCAGCCGATTTGCCATTT-3’
	R:5’-TCCCTCAGGGGTGTCCTTAG-3’
*GAPDH*	F:5’-GACAACTTTGGCATCGTGGA-3’
	R:5’-ATGCAGGGATGATGTTCTGG-3’

## Data Availability

Not applicable.

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
