# Peer review of "Around-the-Clock Noise Induces AD-like Neuropathology by Disrupting Autophagy Flux Homeostasis"

_cells, 2022, doi:10.3390/cells11172742_

Round 1

Reviewer 1 Report

The authors in manuscript entitled “Around-the-clock noise induces AD-like neuropathology by disrupting autophagy flux homeostasis” have studied that mechanism of AD-like changes which caused by exposure to around-the-clock noise and also explored autophagy flux at different stages to acquire a better understanding of the underlying pathological mechanisms.

Strengths of the study:

- This manuscript has written in descriptive manner.

- The reference list is updated, but there is need to add some more references.

- The research article described that this study highlights the exposure to around-the-clock noise that disrupting the homeostasis of autophagy flux, aggravates amyloid and tau pathology in AD-related brain regions as well as causes lysosomal degradation. As the result, Aps accumulate in the cortex and hippocampus in a manner resembling AD-like neuropathology.

There are some issues with this article, if these issues are going to resolve then the quality of the paper is suitable for publication.

1)             In a part of the introduction, it should be crisp and clear about the focused study.

2)             Some recent references should be included.

3)             There are few typos and English and grammar errors which should be rectify.

4)             Figure quality must be improved, Figure is blur and not as per quality of the Journal.

5)             Discussion part must be improved, elaborative and crisp.

6)             Conclusion prospective must be written crisp and clear.

Author Response

Point 1:   In a part of the introduction, it should be crisp and clear about the focused study.

Response 1: Thank you for your careful review. We agree with the reviewer. In the revised manuscript, we have made corrections in the Introduction section as per the reviewer’s comments (Pages 1-2, lines 31-66).

Point 2:  Some recent references should be included.

Response 2: Thank you for this thoughtful suggestion. We agree with the reviewer and have included many recent references in the revised manuscript.

Point 3:  There are few typos and English and grammar errors which should be rectify.

Response 3: Thank you for your careful review. We apologize for the language issues that may have caused any confusion. We have carefully checked the manuscript, making grammar and spelling corrections. The revised manuscript was also edited for language by a professional language editing company one more time.

Point 4:  Figure quality must be improved, Figure is blur and not as per quality of the Journal.

Response 4: Thank you for your comment. The quality of the figures has been improved as suggested.

Point 5:  Discussion part must be improved, elaborative and crisp.

Response 5: We agree with the reviewer and thank them for this thoughtful suggestion. The Discussion section has been revised as suggested (Pages 8-10, lines 230-306).

Point 6:  Conclusion prospective must be written crisp and clear.

Response 6: Thank you for your comment. We agree with the reviewer. Accordingly, we have revised the Conclusion section as per the reviewer’s suggestion (Page 10, lines 308-321).

Reviewer 2 Report

The manuscript by Pengfang Z. et al. interestingly investigates the molecular basis of the AD-like neuropathological signs observed in animal models of hazardous noise. Although the manuscript objectives and hypothesis are strongly validated by previous works of the authors, there are some parts specially of the methodology that need to be clarified as follows:

1.    The authors should provide a statement in the methods that they followed the institutional guidelines and the three R’s principle at the time to plan their animal experiments. Furthermore, a statement that ethical reduction of animal suffering was primordially implemented, in case that it was, should also be clearly stated.

2.    Instead of abundant, maybe the authors should state that the animals were fed ad-libitum.

3.     It would be interesting to know if the effects of the noise have any effect on the auditive system of the animals. Do the authors have analyzed this in their study or any previous study? If so, please state this in the discussion section of the manuscript.

4.    Please, explain better in the statistical section why did you use: ‘One-way analysis of variance followed by Student’s t-test’, a combination that as stated in the methods results as highly confusing.

5.    The authors should cite others recent works in which abnormal authopagy and mitophagy in AD progression has been postulated, such as, PMCID: PMC7210691

6.    English language editing, although is quite well used in the manuscript, should be carefully read to detect some typos.

Author Response

Point 1:   The authors should provide a statement in the methods that they followed the institutional guidelines and the three R’s principle at the time to plan their animal experiments. Furthermore, a statement that ethical reduction of animal suffering was primordially implemented, in case that it was, should also be clearly stated.

Response 1: Thank you for this precious advice. The experimental procedures followed the guidelines specified by the Animal and Human Use in Research Committee of the Tianjin Institute of Health and Environmental Medicine. The Materials and Methods section has been revised as advised (Page 2, lines 74-77).

Point 2:   Instead of abundant, maybe the authors should state that the animals were fed ad-libitum.

Response 2: We agree with the reviewer and thank them for this thoughtful suggestion. We have modified the description as per the reviewer’s comment (Page 2, lines 73-74).

Point 3:  It would be interesting to know if the effects of the noise have any effect on the auditive system of the animals. Do the authors have analyzed this in their study or any previous study? If so, please state this in the discussion section of the manuscript.

Response 3: Thank you for your comment. Our team has been engaged in research on auditory and non-auditory effects of noise for a long time. We have stated the effect of noise on the auditory system based on our previous animal study on Page 9, lines 239-241.

Point 4:  Please, explain better in the statistical section why did you use: ‘One-way analysis of variance followed by Student’s t-test’, a combination that as stated in the methods results as highly confusing.

Response 4: We thank the reviewer for pointing out this writing mistake, which was corrected as suggested (Page 3, lines 127-128).

Point 5:  The authors should cite others recent works in which abnormal authopagy and mitophagy in AD progression has been postulated, such as, PMCID: PMC7210691.

Response 5: We agree with the reviewer and thank them for this valuable suggestion. We have added recent publications on abnormal autophagy and mitophagy in AD progression.

Point 6:  English language editing, although is quite well used in the manuscript, should be carefully read to detect some typos.

Response 6: Thank you for your careful review. We apologize for the language issues that may have caused any confusion. We have carefully checked the manuscript, making grammar and spelling corrections. The revised manuscript was also edited for language by a professional language editing company one more time.

Round 2

Reviewer 2 Report

The authors have successfully addressed all my previous concerns, the paper can be published.